# Equipped for Sexual Stings? Male-Specific Venom Peptides in *Euscorpius italicus*

**DOI:** 10.3390/ijms231911020

**Published:** 2022-09-20

**Authors:** Jonas Krämer, Ricardo Pommerening, Reinhard Predel

**Affiliations:** Institute of Zoology, University of Cologne, Zuelpicher Strasse 47b, 50674 Cologne, Germany

**Keywords:** *Euscorpius*, intraspecific variation, sex specificity, venom composition, proteomic and transcriptomic analysis, toxin

## Abstract

In the animal kingdom, intraspecific variation occurs, for example, between populations, different life stages, and sexes. For venomous animals, this can involve differences in their venom composition. In cases where venom is utilized in the context of mating, the differences in composition might be driven by sexual selection. In this regard, the genus *Euscorpius* is a promising group for further research, as some of these scorpions exhibit a distinct sexual dimorphism and are known to perform a sexual sting during mating. However, the venom composition of this genus remains largely unexplored. Here, we demonstrate that *Euscorpius italicus* exhibits a male-specific venom composition, and we identify a large fraction of the substances involved. The sex specificity of venom peptides was first determined by analyzing the presence/absence patterns of ion signals in MALDI-TOF mass spectra of venom samples from both sexes and juveniles. Subsequently, a proteo-transcriptomic analysis provided sequence information on the relevant venom peptides and their corresponding precursors. As a result, we show that several potential toxin precursors are down-regulated in male venom glands, possibly to reduce toxic effects caused to females during the sexual sting. We have identified the precursor of one of the most prominent male-specific venom peptides, which may be an ideal candidate for activity tests in future studies. In addition to the description of male-specific features in the venom of *E. italicus*, this study also includes a general survey of venom precursors in this species.

## 1. Introduction

Intraspecific variation in venom composition is a widespread phenomenon, observed throughout the animal kingdom [1]. Diet can be an important factor in this variation, which has been demonstrated, e.g., for snakes [2].

For clinically relevant species, studying this variation is crucial to ensure the development of effective antidotes [3,4]. Apart from this, understanding the variation within a species can help to identify substances with specific functions, which might be overlooked otherwise. In this regard, it was shown that sex-specific differences in venom composition can have a strong effect on their biochemical, insecticidal and neurotoxic properties [5]. Hence, it may be useful to also consider intraspecific venom variation when searching for compounds suitable for the development of specific pharmaceutics or pesticides.

Sex-specific differences in venom composition have been demonstrated for several arthropod taxa such as spiders, scorpions, or centipedes [6,7,8]. Different prey preferences due to sexual dimorphisms (e.g., size differences) might be a crucial evolutionary driver of these variations. In terms of understanding the functional purpose of sex-specific venom compositions, spiders are among the best studied groups. As has been demonstrated for the Sydney funnel web spider, the sex specificity of the venom composition can have drastic effects on its efficacy; compounds lethal to humans are only present in the venom of males [9]. Evolutionarily, this might be explained by the development of behavioral differences between the sexes, as males cover long distances in search of females and are thus much more exposed to predators. A reverse situation was discovered for *Loxoceles* spiders, for which females are more lethal to humans [10]. It is also suspected for some arthropods that sex-specific compounds are involved in mating-related functions. An example are spiders of the *Tetragnatha* group, in which courtship behavior involves interlocking of their chelicera [11]. Possibly, the venom is used for communication in this case [12].

Much less is known about sex specificity of scorpion venoms than that of spiders. Even though sex-specific differences in venom composition have been postulated for several scorpion species [8,13,14], only a few compounds involved were identified [15,16]. As some male scorpions perform a sting during mating (‘sexual sting’), the compounds transferred might have a direct relevance for mating. For instance, Olguín-Pérez et al. [14] confirmed that venom transference takes place during the sexual sting of *Megacormus gertschi* and revealed sex-specific differences in the venom composition. They hypothesized that the male-specific compounds function to sedate the female to reduce the risk of cannibalism.

To extend the knowledge about sex-specific venoms in scorpions, the genus *Euscorpius*, which is completely harmless to humans, appears very promising. Members of this genus, which occurs mainly in Europe, often show a pronounced sexual dimorphism; and for some species of this group, the execution of a sexual sting has been demonstrated [17]. For *Euscorpius alpha*, in addition to distinct sex-specific differences in the size and morphology of the telson, differences in the structure of the venom glands were described [18]. However, scorpions of this genus are generally still largely unexplored in terms of their venom composition.

In our study, we performed a comprehensive analysis of the venom composition of *Euscorpius italicus* (Scorpiones: Euscorpiidae), with a focus on identifying sex-specific peptides and the genes encoding them. Our expectation of finding sex-specific venom compounds is based on the previously described morphological differences of the venom delivery system between the sexes of this genus.

## 2. Results

The sexual dimorphism previously described for *E*. *alpha* was also observed in *E. italicus* (Figure 1). The telsi of males are inflated and considerably larger than those of females (Figure 1a). In addition, the tip of the stinger appears to be more curved. The morphology of the telson of last stage juveniles resembles that of females. Figure 1c,d document this transition of telson morphology from juvenile to male *E. italicus*. As venom was collected from marked specimens before and after the adult molt, we were also able to document the transition in venom composition from juveniles to males in these individuals (see below). The transition not only affected the venom composition, but also the amount and consistency of venom released during electrical stimulation (Figure 2). Basically, the quantity of venom released increased significantly in males (Figure 2a, Appendix A) and the venom became translucent (Figure 2b).

### 2.1. MALDI-TOF Mass Spectra Reveal a Male-Specific Ion Signal Pattern

We analyzed MALDI-TOF mass spectra (hereafter referred to as mass fingerprints) from repeated venom extractions of six adults (two males, four females) and four last instar juveniles (two juvenile males, two juvenile females). Representative mass fingerprints (m/z 3000–10,000) are shown in Figure 3. No substantial differences were observed in venom mass fingerprints from females and juveniles. However, mass fingerprints of venom released from males appeared to systematically differ from those of juveniles and females (Appendix A). Especially in the mass range of m/z 3000–10,000, several prominent ion signals typical of mass fingerprints from females/juveniles were not detected. On the other hand, mass fingerprints from the venom of males show additional signals that were not observed in mass fingerprints of venom from females/juveniles. In Figure 3, these signals are highlighted. We also analyzed the transition in venom composition from juveniles to males using the same individuals before and after the adult molt (Appendix A).

### 2.2. Combined Proteo-Transcriptomic Analysis with a Focus on Identifying the Male-Specific Changes of the Venom

This study comprises a comprehensive venom analysis of male and female *E. italicus* with a focus on identifying products of genes with sex-specific expression. The basic information for assigning sex-specific compounds and their respective genes was obtained by repetitive MALDI-TOF mass spectrometry (MS) analyses of venom samples (see above). These experiments showed that in adult males the venom composition changed dramatically, while in females the venom composition was largely identical to that of juvenile scorpions. The changes in the venom of male *E. italicus* suggest both up- and down-regulation of genes.

This information was used to identify the corresponding gene products using a proteo-transcriptomic approach in which proteomic venom analyses using Quadrupole Orbitrap MS were matched with transcriptome data generated from telsi of both sexes with their venom glands and an additional transcriptome generated from tail tissue without telson (negative control). Next-generation sequencing resulted in 18,806,043 paired-end reads for the female telson transcriptome, 20,946,933 paired end reads for the male telson transcriptome, and 22,576,333 paired end reads for the negative control, each after removal of adapter sequences. The corresponding assemblies contain 85,185 contigs for the female telson transcriptome, 66,778 contigs for the male telson transcriptome, and 80,811 contigs for the negative control transcriptome. The assessment of transcriptome completeness revealed 86.9% complete BUSCOs and 3.2% fragmented BUSCOs for the male telson transcriptome, 87.2% complete BUSCOs and 3.2% fragmented BUSCOs for the female telson transcriptome, and 94.1% complete BUSCOs and 1.0% fragmented BUSCOs for the negative control transcriptome.

For venom samples of both sexes, proteome data of two Quadrupole Orbitrap MS experiments (one experiment with trypsin digestion and one without) were matched to the transcriptome assemblies. After quality filtering, removal of precursors without signal peptide and redundant matches, 326 precursors were identified that contribute to the venom composition of both sexes of *E. italicus*. The complete list of venom precursors can be found in Appendix A. Based on similarity to sequences from the UniProt database, these were classified as precursors of potential neurotoxins, enzymes, antimicrobial peptides (AMPs), others and, for no matches with the database, as novel (Figure 4). Figure 4 also illustrates the percentage of precursors for which up- or down-regulation in males could be confirmed in the present study. Details of these precursors, including the most similar Blast hits in the UniProt database, are shown in Table 1. All in all, eleven precursors were identified, with the majority down-regulated in males. Although the corresponding mature peptides could not be detected in MALDI-TOF mass spectra of venom samples from males and their precursors show much higher expression levels in the telson transcriptome of females, the expression level in male telson transcriptome was still higher compared to the negative control. Consistent with this, the proteomic data from Quadrupole Orbitrap MS also revealed low amounts of these peptides in male venom samples, indicating significant quantitative differences rather than presence/absence. Most of these precursors, which are down-regulated only in males, process cysteine-rich peptides, classified here as potential neurotoxins based on their similarity to corresponding scorpion sequences (Table 1). Of these, EUTX-Ei2a/EUTX-Ei2b and EUTX-Ei3a/EUTX-Ei3b are likely products of paralogous genes. For EUTX-Ei2b, we classify the peptide as a potential neurotoxin, although the most similar BLAST hit at UniProt lists an AMP of the scorpion *Hadrurus spadix* (Table 1). This classification is based on the presumed relationship of EUTX-Ei2b to EUTX-Ei2a, the latter showing in turn a high similarity to a sodium channel toxin of the scorpion *M. gertschi*. Regarding the precursor processing, the potential mature neurotoxins either correspond to the complete precursor sequences without signal peptide (EUTX-Ei1, EUTX-Ei3a), use the C-terminal Gly as amidation signal (EUTX-Ei3b), or the C-termini result from cleavage at a C-terminal tribasic cleavage site (EUTX-Ei2a, EUTX-Ei2b).

Three precursors down-regulated in males are characterized by the absence of cysteines and are provisionally assigned here as AMP precursors (AMPs-Ei1, 2a, 2b). Their mature peptides are short and located in the precursor between the signal peptide and a monobasic (AMP-Ei1) or dibasic cleavage site (AMP-Ei2a, 2b), the former as part of a quadruplet motif [20]. The peptides of the likely paralogous AMP-Ei2a and AMP-Ei2b genes are amidated and thus better protected against degradation. Their C-terminal sequence resembles that of insect SIFamides, which are described as neuropeptides [21]. Processing of all putative AMP precursors potentially also releases one or more short peptides located C-terminally from the aforementioned cleavage sites, but these products were not detected by MS. One of the precursors down-regulated in males and yielding no significant BLAST hits in the UniProt database was classified as novel (Novel Venom Compound Ei1). The corresponding mature peptide with two cystines (C-C) results from cleavage at an N-terminal quadruplet motif, the C-terminal amino acid in the precursor sequence functions as amidation signal (Table 1).

Three precursors were identified as up-regulated in males, two of which were classified as potential neurotoxins based on the BLAST hits in the UniProt database (Table 1). With EUTX-Ei4, we identified the precursor associated with the most prominent male-specific MALDI-TOF ion signal in terms of intensity. The corresponding mature peptide with three cystines is the complete precursor without signal peptide, the C-terminal Gly is used as amidation signal. The much less abundant EUTX-Ei5 also results in a mature peptide comprising the complete precursor without signal peptide but does not have a C-terminal amidation. The third precursor, up-regulated in males, is classified as precursor of a cysteine-rich protease inhibitor. The mature peptide is C-terminally amidated, its sequence in the precursor is between the signal peptide and a short C-terminal precursor peptide that is N-terminally cleaved at a monobasic cleavage site (Arg).

## 3. Discussion

Using a combination of proteomics and transcriptomics, we performed the first comprehensive venom analysis for a scorpion of the species-rich genus *Euscorpius*. In addition to the general goal of assessing the complement of genes involved in venom production, our special focus was on analyzing the presumed sexual dimorphism of the venom of *Euscorpius* at a molecular level. A marked dimorphism on the morphological level was previously described for *E. alpha* [18] and comprises much larger male telsi and sex-specific differences in the cell architecture of the venom glands. For *E. italicus*, a similar sexual dimorphism, at least with respect to the telson morphology, has already been described [22] and was also observed by us for the specimens used in the current study. That this dimorphism also relates to an altered cellular composition of the venom gland seems plausible, as the male venom released after electrical stimulation has a much higher quantity and transparency. The transparent nature of the venom we observed in male *E. italicus* resembles the ‘prevenom’ described for *Parabuthus transvaalicus* (Buthidae), which has a lower protein concentration [23]. For *E*. *alpha*, it has been shown that the male venom glands contain much less of the granule-rich glandular cells that are assumed to produce a more concentrated venom [18]. In our search for sex-specific compounds in the venom of *E. italicus*, we found that only the composition of male venom changes after adult molt. This corresponds to the observed changes in telson morphology of males. In females, apart from size, there are no morphological changes compared to the juvenile stages. Thus, the sex specificity in the venom of *E. italicus* is in fact male specificity. No reproducible differences in the venom composition of females and juvenile scorpions could be found, regardless of whether the juveniles developed into females or males.

Using MALDI-TOF MS, we could conclusively demonstrate that the transparent male venom of *E*. *italicus* is indeed less complex in terms of peptides. In our subsequent structural elucidations, we focused exclusively on those precursors whose peptide ion signals were clearly different in MALDI-TOF mass spectra from those of females or juveniles. In our approach, we considered the peptides enriched in the venom with a molecular mass of up to 10,000 Da; other substance groups were not included. In the chosen mass range, we were able to identify most precursors of those venom peptides that are either up- or down-regulated in male venom. As verified by Q-Exactive MS and transcriptomics, up- and down-regulation of the respective genes is essentially a quantitative phenomenon. Small amounts of the peptides not detected by MALDI-TOF MS in venom samples from males or juveniles/females are nevertheless always present in both sexes. Similarly, Binford et al. [24] found relative differences in abundance of venom precursors for the spider *Tegenaria agrestis*. It is possible that sex-specific differences in venom composition observed in the past will also turn out to be merely quantitative differences when more sensitive analytical approaches are applied.

The evolutionary driver of a male-specific venom composition in *E. italicus* could be sexual selection [25]. For spiders, it was initially assumed that chemical differences in venom composition result from different feeding preferences of the sexes [26], which was corroborated by frequently observed sexual dimorphisms [27,28]. However, when Binford et al. [11] wanted to confirm this hypothesis for spiders belonging to the genus *Tetragnatha*, this correlation could not be confirmed. Interestingly, these spiders perform a special courtship behavior in which the chelicera are interlocked. Venom transfer during this procedure is likely, and a function of sex-specific venom compounds in mating is therefore hypothesized [11]. A special courtship behavior in the form of a sexual sting during mating has also been described for *Euscorpius* [17]. This behavior is performed as part of the ritualized mating dance, known as promenade a deux [29], and has meanwhile been observed in several scorpion families [30]. Compared to a predatory/defensive sting, the duration of the sexual sting is extraordinarily long, ranging from few minutes [31] up to 50 min [14]. That the sexual sting involves venom injection into the female was confirmed at least for a Mexican scorpion, which also belongs to the family Euscorpiidae [14]. Given the more transparent male venom of that species, it has been hypothesized that venom injected during sexual sting might contain less toxins to prevent harm to females [23]. Our findings support this assumption, as several of the compounds down-regulated in male venom were classified as potential neurotoxins.

However, as shown here, male venom of *E. italicus* is not simply a diluted solution with less toxins, but a few genes are highly up-regulated, leading to a male-specific enrichment of their mature products in the venom. This enrichment strongly suggests a utilization in a mating-related context. Regarding possible functions of these compounds, several ideas are plausible. Their use could solely have a beneficial effect on the male fitness, e.g., by sedating females to facilitate sperm transfer and reduce the risk of cannibalism [32]. On the other hand, females could benefit, e.g., by assessing male suitability based on the injected compounds [14]. Finally, injection of male-specific compounds might promote fertilization or help relax female musculature to facilitate the mating procedure [23]. In *E*. *italicus*, the most prominent male-specific venom peptide (EUTX-Ei4) shows similarity to a sodium channel toxin. To unravel the function of such compounds, which may be linked to the sexual sting, future studies should address the effects caused by their injection in more detail. In a first step, male or female crude venom could be injected into females of *E. italicus*; in further experiments synthetic venom peptides could then be used to decipher their specific function(s). It appears reasonable that venom peptides down-regulated in males could be harmful to females, while venom peptides up-regulated in males cause changes in female behavior. Based on the observed down-regulation of several putative neurotoxin genes in male venom glands, also a reduction in venom toxicity for prey might be assumed. Male *E*. *italicus* do eat and use their stinger to subdue prey (personal observation). A reduced venom toxicity might be compensated by the relatively large male chelae though. In future studies, it might be interesting to test for behavioral difference during prey capture (e.g., frequency of stinging) and to compare male and female venom toxicity to prey.

## 4. Materials and Methods

### 4.1. Acquisition and Rearing of the Animals

Ten specimens of *E. italicus* were obtained from an online shop (https://www.vogelspinnen.shop/, accessed on 24 November 2020). These were offspring from specimens originating from Slovenia. Originally, two of these animals were males, while the rest were either females or juveniles. During this study, several juveniles had their adult molt. Two became males, one became female and the remaining juvenile was sacrificed for RNA extraction before the final molt. The scorpions were kept individually in plastic boxes equipped with coconut substrate and a piece of wood as a hideout. Once a week, they were fed with *Acheta* nymphs and water was provided to ensure sufficient moisture. In case the animals were used for venom extraction, feeding was performed immediately afterwards.

### 4.2. Sex Determination of the Animals

For males, determination of sex and maturity was unambiguous due to the presence of several male-specific morphological traits (telson, pectine organ, and chelal hand [22,33]). For the remaining specimens (including the one used for RNA extraction), females were identified based on missing male-specific traits. This also enabled differentiating the sex of subadults, as some male-specific traits can already be observed in subadults (male-specific characteristics of pectine organ, presence of genital papillae). Differentiation from instars was mainly based on size (specimens exceeding a body length of 20 mm without metasoma were regarded as adults). As the size alone is a weak criterion for estimating maturity, at least some specimens classified as mature might have been subadult. At least for two females, observations of mating behavior supported their maturity (Appendix A); instars and unreceptive females successfully fend off male mating attempts [34].

### 4.3. Venom Collection

To extract the venom, the scorpions were immobilized with rubber bands using the device shown in Figure 5. The release of venom was triggered by applying electricity (12 V, 250 µs pulse width, and 150 Hz pulse rate) to the telson integument using a promed tens device (Promed GmbH, Farchant, Germany) equipped with modified tweezers [35]. Conduct gel was used to improve electrical current flow. Venom was collected into glass capillaries (inner diameter 1 mm; Hilgenberg GmbH, Malsfeld, Germany) and transferred in 20 µL of Millipore water. Storage of venom samples was at −20 °C.

### 4.4. Quadrupole Orbitrap MS with Nanoflow HPLC

For each sex, two Quadrupole Orbitrap MS experiments were performed, one bottom-up analyses with digested samples and the other without digestion step, but with reduction/alkylation of the sample. For the top-down experiments, venom samples were mixed with an equal volume of urea buffer (8 M urea/ 50 mM triethylammonium bicarbonate buffer) for denaturation prior to reduction/alkylation. For desalting and removal of urea, poly (styrene divinylbenzene) reverse phase (RP)-StageTip purification was performed before Orbitrap MS analyses according to the StageTip purification protocol from the CECAD Proteomics Facility, University of Cologne (http://proteomics.cecad-labs.uni-koeln.de/Protocols.955.0.html, accessed on 15 February 2021). The protein concentration of venom samples (diluted in 20 µL Millipore water) for bottom-up analyses was measured with Direct Detect^TM^ Spectrometer (Merck, Germany), which resulted in a final concentration of 4.935 µg/µL for the diluted male venom and 2.372 µg/µL for the diluted female venom used for the Quadrupole Orbitrap MS. Digestion of venom samples for the bottom-up analyses was performed according to the in solution digestion protocol (http://proteomics.cecad-labs.uni-koeln.de/Protocols.955.0.html, accessed on 15 February 2021). For both sexes, 25 µg of venom was used. Trypsin/LysC digestion was performed with 0.5 μg trypsin and 0.5 μg LysC. Afterwards, the venom compounds/tryptic peptides were separated on an EASY nanoLC 1000 UPLC system (Thermo Fisher Scientific, Bremen, Germany). For this purpose, inhouse packed RPC18 columns with a length of 50 cm were used (fused silica tube with ID 50 μm ± 3 μm, OD 150 μm; Reprosil 1.9 μm, pore diameter 60 A°; Dr. Maisch GmbH, Ammerbuch-Entringen, Germany). UPLC separation was performed with a binary buffer system (A: 0.1% formic acid (FA), B: 80% acetonitrile, 0.1% FA): linear gradient from 2 to 62% in 110 min, 62–75% in 30 min, and final washing from 75 to 95% in 6 min (flow rate 250 nL/min). Re-equilibration was performed with 4% B for 4 min. The UPLC was coupled to a Q-Exactive Plus (Thermo Fisher Scientific) mass spectrometer. HCD fragmentations were performed for the 10 most abundant ion signals from each survey scan in a mass range of m/z 300–3000. The resolution for full MS1 acquisition was set to 70,000 with automatic gain control target (AGC target) at 3 × 10^6^ and a maximum injection time of 80 ms. In order to obtain the HCD spectra, the run was performed at a resolution of 35,000, AGC target at 3 × 10^6^, a maximum injection time of 240 ms, and 28 eV normalized collision energy; dynamic exclusion was set to 25 s.

### 4.5. MALDI-TOF MS

MALDI-TOF mass spectra were generated for two venom samples extracted from each of the ten specimens with a one-week interval. Venom samples used for MALDI-TOF MS were diluted in a mixture of ethanol and TFA (final concentration: 35%/0.1%) to achieve optimal analyte/matrix ratios. As reference the volume of extracted crude venom was used. Optimal dilutions were tested beforehand in dilution series performed with venom from both sexes and juveniles. Subsequently, 0.3 μl was directly spotted onto the sample plate and mixed with the same volume of 10 mg/ml 2.5-dihydroxybenzoic acid (Sigma Aldrich, Steinheim, Germany) matrix, dissolved in 50% acetonitrile/0.05% TFA. For an optimal crystallization of the matrix, samples were blow-dried with a hairdryer. An ultrafleXtreme TOF/TOF mass spectrometer (Bruker Daltonik GmbH, Bremen, Germany) was used in reflectron positive mode with overlapping mass ranges of m/z 800–4500 and m/z 3000–10,000. For an optimal signal-to-noise ratio, laser intensity and the number of laser shots were adjusted for each sample. Laser frequency was set to 666 Hz. For external calibration, a mixture containing proctolin ([M+H]^+^, 649.3), Drm-sNPF-21^2–19^, ([M+H]^+^, 974.5), Pea-FMRFa-12 ([M+H]^+^, 1009.5), Lom-PVK ([M+H]^+^, 1104.6), Mas-allatotropin ([M+H]^+^, 1486.7), Drm- IPNa ([M+H]^+^, 1653.9), Pea-SKN ([M+H]^+^, 2010.9), and glucagon ([M+H]+, 3481.6) was used for the mass range of m/z 800–4500 and a mixture of bovine insulin ([M+H]^+^, 5731.5), glucagon and ubiquitin ([M+H]^+^, 8560.6) was used for the mass range of m/z 3000–10,000. Ion signals were identified by using the peak detection algorithm SNAP from the flexAnalysis 3.4 software package. In addition, each spectrum was manually checked to ensure that the monoisotopic peaks were correctly identified.

### 4.6. RNA Extraction, Transcriptome Sequencing and De Novo Assembly of Nucleotide Sequences

For both sexes, one transcriptome was generated for the telson comprising the venom glands. As a negative control, one transcriptome was generated for parts of the scorpion tail without the venom glands. RNA extraction was performed with TRIzol (Thermo Fisher Scientific, Darmstadt, Germany) according to the standard manufacturer’s user guide. Libraries were prepared starting from 1 μg of total RNA with the Illumina^®^ TruSeq^®^ stranded RNA sample preparation Kit. Vallidation and quantification of libraries were performed with the Agilent 2100 Bioanalyzer. Paired-end sequencing was performed using an Illumina TruSeq PE Cluster Kit v3 and an Illumina TruSeq SBS Kit v3—HS on an Illumina HiSeq 4000 sequencer. Adapter removal and quality trimming of resulting raw files was performed using Trimmomatic 0.3.2 [36]. Afterwards, de novo assembly of RNA sequence data was performed with Trinity v2.8.5 [37] based on default settings. This assembly was used to search for peptide sequences obtained by Quadrupole Orbitrap and MALDI-TOF MSMS experiments. To assess the completeness of transcriptomic data, BUSCO 4.1.4 [38] was used. Transcriptomic data were submitted to the National Center for Biotechnology Information (NCBI, Bioproject: PRJNA877843).

### 4.7. Identification of Venom Compounds

Identification of venom compounds was performed similarly as described in [39]. Precursors of potential venom compounds were identified by matching the fragment spectra of Quadrupole Orbitrap MS analyses against the telson-transcriptomes of male and female *E. italicus*, utilizing the software PEAKS 10 (PEAKS Studio 10; BSI, Toronto, ON, Canada). Then, precursors were characterized by predicting a signal peptide with SignalP 6.0 [40], checking for the presence of a stop codon, searching for respective precursors in the negative control (BLAST search against the remaining telson transcriptome with an E-value of 1 × 10^−5^) and estimating the expression level in transcriptomic data with Kallisto [41]. Additionally, the precursors were classified based on similarities to compounds from the online database UniProt [19], by performing local BLAST searches in the Metazoa database (search term: ‘taxonomy:”Metazoa [33, 208]’) and the Tox-Prot database (search term ‘taxonomy:”Metazoa [33, 208]” (keyword:toxin OR annotation:(type:”tissue specificity” venom))). As E-values, 1 × 10^−5^ was used for the search in the Metazoa database and 0.1 for the Tox-Prot database. Classification of the matches was performed based on the description of the BLAST hits, for the Tox-Prot database only in case the E-value was lower than 1 × 10^−5^. Finally, the matches were filtered with respect to the quality of MS data, the coverage between transcriptomic and proteomic data (false discovery rate (−10lgP) > 30, coverage > 7%) and the presence of a signal peptide. To perform a functional annotation all identified venom precursors were analyzed with InterProScan [42].

To verify the presence of the identified venom precursors in the venom, theoretical masses calculated for each of the potential bioactive peptides (predicted from the identified venom precursors) were searched against a list of ion signals from MALDI-TOF mass spectra of the venom.

The presence of corresponding ion signals in mass fingerprints of the venom was also utilized to assess the sex specificity of venom compounds. For this estimation, only highly reproducible MALDI-TOF ion signals were considered (occurrence in at least 50% of the spectra). This assessment was based on the generation of repeated MALDI-TOF mass spectra for two adult males, four adult females and four juveniles.

## Figures and Tables

**Figure 1 ijms-23-11020-f001:**
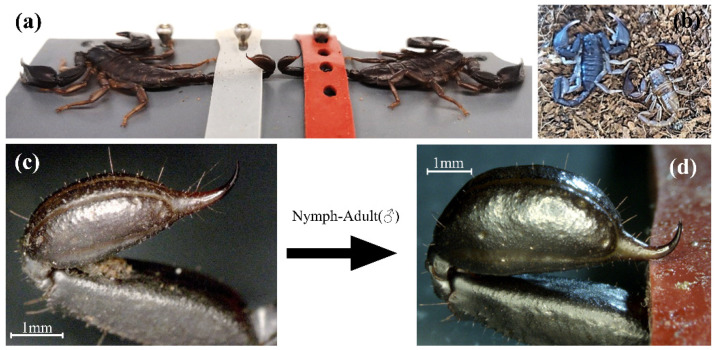
External telson morphology of *E*. *italicus*. (**a**) Comparison of male (left) and female (right) specimens with differently shaped telsi. (**b**) Male after adult molt with last instar exuvia. (**c**,**d**) Telson morphology of a single individual before (**c**) and after (**d**) molting into a male.

**Figure 2 ijms-23-11020-f002:**
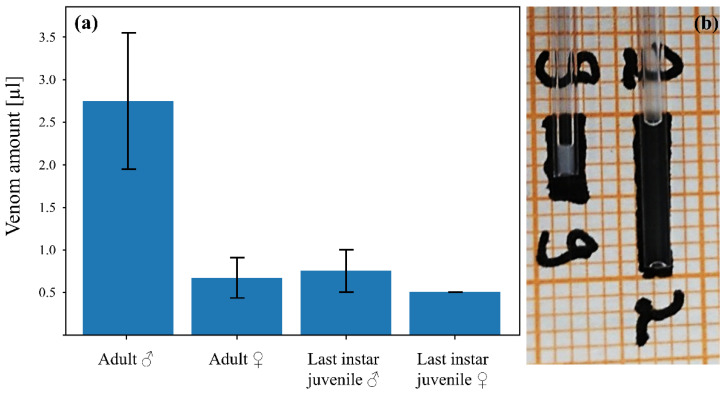
Comparison of venom amount and consistency of the sexes of *E*. *italicus*. (**a**) Mean venom amount (error bars represent standard deviation) released by adult males (6 extractions; 2 specimens), adult females (12 extractions; 4 specimens), and juveniles (12 extractions; 4 specimens) during venom extraction. The values are based on three repeated extractions per specimen with an interval of one week between the extractions. (**b**) Venom consistency in males (right capillary; translucent) and females (left capillary; milky).

**Figure 3 ijms-23-11020-f003:**
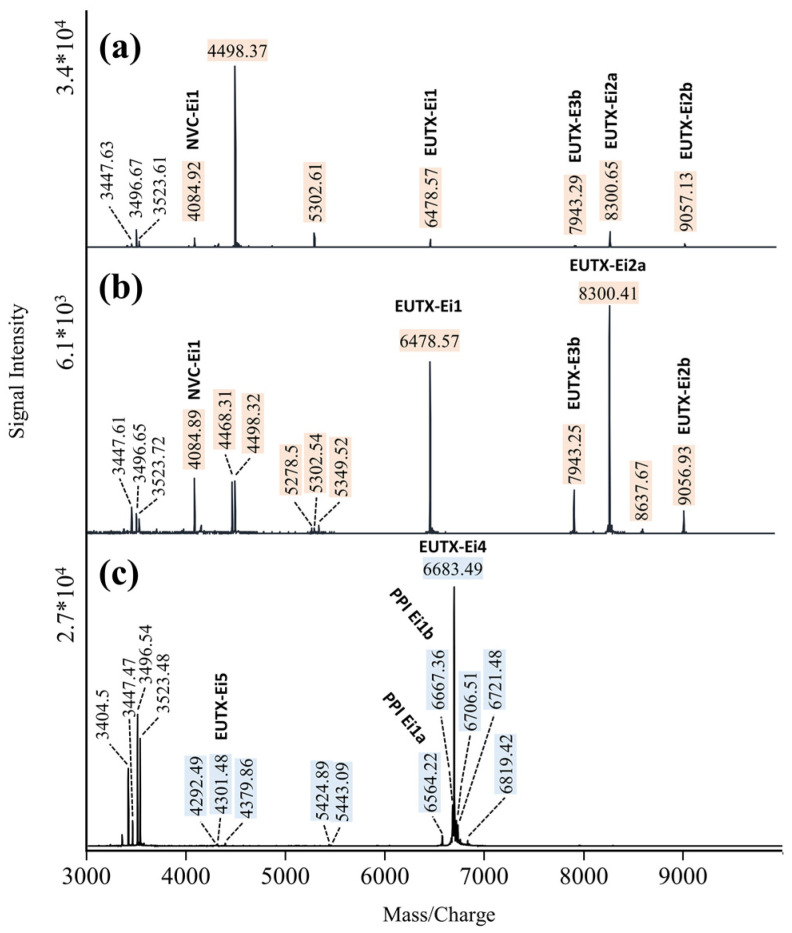
Comparison of MALDI-TOF mass fingerprints (m/z 3000–10,000) of venom extracted from juvenile (**a**), female (**b**), and male *E. italicus* (**c**). Male-specific ion signals are highlighted in blue; ion signals that are missing in mass spectra of venom from males are highlighted in beige.

**Figure 4 ijms-23-11020-f004:**
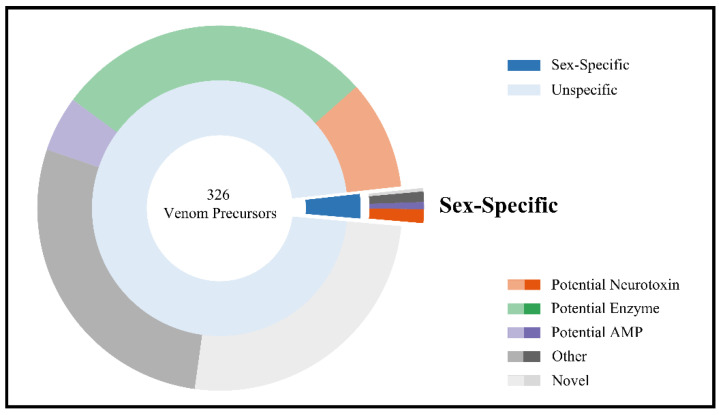
Pie chart highlighting the fraction of venom precursors that were confirmed to be down- or up-regulated (“sex-specific”) in males of *Euscorpius italicus* (inner circle). In the outer circle, venom compounds are classified based on similarities to sequences from UniProt [19] entries.

**Figure 5 ijms-23-11020-f005:**
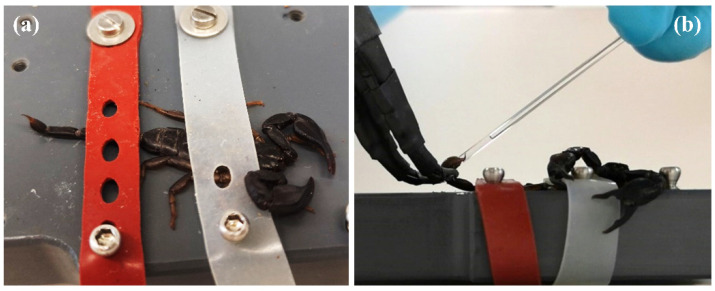
Venom extraction from *E. italicus*. (**a**) Fixation of specimens with rubber tubes. (**b**) Application of electricity with modified pincers and collection of released venom into glass capillaries.

**Table 1 ijms-23-11020-t001:** Information on venom precursors up- [↑] or down-regulated [↓] in males of *E. italicus*. Specificity was assessed based primarily on repeatedly generated MALDI-TOF mass spectra of venom samples from both sexes and juveniles but was also supported by the statistically less significant transcriptomic and Quadrupole Orbitrap MS experiments. Precursors were identified by a combined proteomic and transcriptomic approach. Grey, signal peptide; blue letters, potential bioactive peptide; green and italic, amidation signal; yellow, cysteine (half of the disulfide bond); red letters, cleavage site. Black underlined, confirmed by MSMS; red underlined, confirmed only by MSMS of digested samples; dashed line, mass match in MALDI-TOF MS. The column Orbitrap MS indicates which of the four analyses confirmed the precursor sequence. D, digested samples; ND, samples without digestion; m, venom from males; f, venom from females. Assumed PTMs include amidation (A) and disulfide bridges (C–C).

Name	BLAST Hit	Expression Level ♂ [tpm]	Expression Level ♀ [tpm]	Expression Level NegativeControl [tpm]	PTM	Predicted Mass[M+H]^+^	Orbitrap MS	Male Specificity (MALDI)

U-Euscorpiustoxin-Ei1	Putative potassium channel toxin (*M. gertschi*), 63%, Acc: A0A224XGQ4	349.45	4136.68	-	C-C	6478.97	D (m/f), ND (m/f)	↓
MVKQLVAAFLVIMLISSLVDA ** KKTFMEKAKSVFSKAGNKIKEIAGKSEYMCPVVSSFCEQHCARQEKSGECDFNKCTCS **
U-Euscorpiustoxin-Ei2a	Putative sodium channel toxin (*M. gertschi*), 71%, Acc: A0A224XBU0	2771.21	13,995.7	-	C-C	8301.05	D (m/f), ND (m/f)	↓
MNAKLTVLLFLAMVAIASC ** GWINEKRVQSYIDEKIPNGVMKGAIKAVVHKIAKNEYGCVANIDTVSQCNKHCIAAGSEKGVCHGTKCKCDKELSY ** ** RRK **
U-Euscorpiustoxin-Ei2b	Antimicrobial peptide (*Hadrurus spadix*), 66%, Acc: A0A1W7RB12	766.38	6848.29	-	C-C	9056.51	D (m/f), ND (m/f)	↓
MQIRCSILLLLMISSFCSC ** GILREKYFHQAVDKVAPMIPLPVVSQVVGNVAKQIVHKFAKNEALCMFNKDVAGMCDKSCKEAGKSNGICHGTKCKCDKPLSY ** ** KKK **
U-Euscorpiustoxin-Ei3a	La1-like protein 15 (*Urodacus yaschenkoi*), 51%, Acc: L0GB04	8058.69	29,307.7	2.73	C-C	8637.12	D (m/f), ND (m/f)	↓
MKRLQVAALVCLLLCALFSLSAG ** AGEICEANGLSIPVGQDKQDPKSCDLYKCIMQNNRLVLDKFS ** ** CATLKR ** ** K ** ** R ** ** GCKIVPGDSKAAFPKCCPTSNCRGAQWDQ **
U-Euscorpiustoxin-Ei3b *	Putative La1-like peptide (*M*. *gertschi*), 68%, Acc: A0A224X3N5	2320.13	18,041.9	3.49	A,C-C	7942.79/7956.88	D (m/f), ND (m/f)	↓
MENALGGVMLGSLLLLSLFSASLA ** IGEKCETGQHVI ** ** D/E ** ** VGKQVQDSKSCTLYKCINYN ** ** R ** ** KYAL ** ** ETLTCASQKL ** ** KSGCR ** ** SIPGAANTPFPN ** ** CCPTVICQ ** * ** G ** *
Antimicrobial Peptide Ei1	Putative non-disulfide bridge peptide (*M. gertschi*), 81%, Acc: A0A224XFL9	2015.04	9050.66	-	-	2586.38	D (m/f), ND (m/f)	↓
MHFNKTLLVIFLSYLLVTDEAEA**FWGFLAKLATKVVPSLFGSSSEKSK****R**EIENFFEPYQKDLDLELDRFDRFLSKLDLN
Antimicrobial Peptide Ei2a	Antimicrobial peptide UyCT3 (*Urodacus yaschenkoi*), 59%, Acc: L0GCI6	1228.17	11,051.7	-	A	1474.82	D (m/f), ND (m/f)-	↓
MKNQFVILIIAVVLLQLFSPSEA**ILSDIWNGIKGLF*****G*****KR**GLFPQRPLINRDQFDDVFDDDLSAADLKFLQELLK
Antimicrobial Peptide Ei2b	Antimicrobial peptide UyCT3 (*Urodacus yaschenkoi*), 65%, Acc: L0GCI6	637.46	2871.02	-	A	1504.83	D (m),ND (m/f)	↓
MKNQFVILVIAVVLLQLFSPSEA**ILSDIWNGIKSIF*****G*****KR**GLRNLDRFDDLFDDDVSDADLKVLQELFR
Novel Venom Compound Ei1 *	-	5.1	3418.27	-	A,C-C	4085.01/4086.00	D (f),ND (m/f)	↓
MKCYLAVLVLLLVCAVLPDQTCGIENG**R****KSPNFCRNKCLKEYIPNNCVGYCERVLKEKE****K/E****KE*****G****
U-Euscorpiustoxin-Ei4	Putative Na+ channel toxin (*Superstitionia donensis*), 50%, Acc: A0A1V1WBR1	8482.11	60.96	17.54	A,C-C	6683.95	D (m/f), ND (m/f)	↑
MKWCTVFMFCLVILVHEFQDVYG ** EKEGYPLDATRNIYQCYDLGENDYCEKKCKEFGGHGYCYGFACYCKYIRDDVKIWK ** * ** G ** *
Putative Protease Inhibitor Ei1a	Chymotrypsin-elastase inhibitor ixodidin-like (*Ixodes scapularis*), 37%, Acc: A0A2R4SV19	1712.29	218.501	88.11	A,C-C	6563.75	D(m)	↑
MTNLRETVANMKTLAVTLLTLAAFQLVLP**YPQPEESSPPENCGENELFYGRRTCAPICNDEVCKKPSREPTACFAICYQGCYCK****EGY*****G*****R**NRRNETCVKCE
U-Euscorpiustoxin-Ei5	AKTx (*Hadrurus spadix*), 35%, Acc: A0A1W7RB23	1812.84	23.48	163.27	C-C	4301.78	D (m),ND (m)	↑
MNLIIIFTLLLSSPFIEVEG ** SQVNARASCTNSGVCRSSTCPSRGC ** ** RSGK ** ** CINRKCTCYYC **

* Allelic difference (relevant amino acids highlighted in green).

## Data Availability

Transcriptomic data generated in this study were submitted to NCBI (Bioproject: PRJNA877843). Transcriptome raw data were submitted to the Sequence Read Archive (Male: SRR21484294, Female: SRR21484302).

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
