# Peer review of "Equipped for Sexual Stings? Male-Specific Venom Peptides in Euscorpius italicus"

_ijms, 2022, doi:10.3390/ijms231911020_

Round 1
Reviewer 1 Report
Nice work overall. I look forward to seeing this in print. Attached are some comments. I hope you find them helpful.

Reviewer 2 Report
The manuscript by Kramer et al described the differences in venom compositions of male vs female scorpion Euscorpius italicus, assessed through mass spectrometry of venoms and RNA sequencing of venom glands. Based on compositions, it appears that a majority of venom toxins are present in all samples regardless of sex. Venom compositions of female and juvenile E. italicus appeared to be similar. However, in male, many toxins were down-regulated while a few other were up-regulated, resulting in venom composition that is relatively lower in complexity. Therefore, the authors concluded that only male venom changes after adult molt. As E. italicus performs sexual sting during mating, it was suggested that the down-regulation of multiple potential neurotoxins in male may have the role of reducing toxicity to female.
Overall, this study explored an interesting subject that may be of interest to readers across several different disciplines. The manuscript was well-written and data were clearly presented. I only have a few minor comments for the authors’ consideration.
1. It is not immediately clear to me how the relative abundance of each proteins were quantified? ie. what was the method used to determine the up- vs down- regulation? Since quantitative proteomics (eg. ITRAQ) were not performed, I assumed it was by transcripts count from RNA-seq? While the manuscript did mentioned “As verified by Q-Exactive mass spectrometry and transcriptomics, up- and downregulation of the respective genes is essentially a quantitative phenomenon” (page 9 line 32), I struggled to find detail descriptions of the process. Can the authors please add one such paragraph in the Materials and Methods section? This is crucial as it directly affects the conclusion of the study.
2. Similarly, although I trust the authors have standardized quantity of proteins across samples from male vs female vs juvenile, I only see the relevant statement with reference to bottom-up proteomics (tryptic digestion) but not top-down & MALDI-TOF. Please include the method of protein quantification as well.
3. While male may have several toxins down-regulated, the volume of venom injected appeared to be higher (Fig 2). Yet it also appeared that male venom is more concentrated (page 11 line 123) – can the authors comment on this, and also how it affects the speculation that down-regulation of toxins can reduce toxicity incurred upon female? Would this effect be compensated by the increase in total volume injected? Also, how does this change in relative abundance of toxins in venom affect their feeding behavior? Would the down-regulation to reduce toxicity for sexual sting confer evolutionary disadvantage to male in terms of feeding? I understand most of these are hypothesis and speculations, nonetheless, more discussion along the line may be interesting.
4. Fig. 4: “neurotoxin” (legend)
